# Tumor-Extrinsic Axl Expression Shapes an Inflammatory Microenvironment Independent of Tumor Cell Promoting Axl Signaling in Hepatocellular Carcinoma

**DOI:** 10.3390/ijms25084202

**Published:** 2024-04-10

**Authors:** Kristina Breitenecker, Denise Heiden, Tobias Demmer, Gerhard Weber, Ana-Maria Primorac, Viola Hedrich, Gregor Ortmayr, Thomas Gruenberger, Patrick Starlinger, Dietmar Herndler-Brandstetter, Iros Barozzi, Wolfgang Mikulits

**Affiliations:** 1Center for Cancer Research, Comprehensive Cancer Center, Medical University of Vienna, 1090 Vienna, Austriadenise.heiden@meduniwien.ac.at (D.H.); tobias.demmer@meduniwien.ac.at (T.D.); gerhard.weber@meduniwien.ac.at (G.W.); viola.hedrich@meduniwien.ac.at (V.H.); gregor.ortmayr@meduniwien.ac.at (G.O.); dietmar.herndler-brandstetter@meduniwien.ac.at (D.H.-B.); iros.barozzi@meduniwien.ac.at (I.B.); 2Department of Surgery, HPB Center, Viennese Health Network, Clinic Favoriten and Sigmund Freud Private University, 1100 Vienna, Austria; 3Department of Surgery, Mayo Clinic, Rochester, MN 55905, USA; 4Centre of Physiology and Pharmacology, Medical University of Vienna, 1090 Vienna, Austria

**Keywords:** hepatocellular carcinoma, Axl, Gas6, tumor microenvironment, inflammation

## Abstract

The activation of the receptor tyrosine kinase Axl by Gas6 is a major driver of tumorigenesis. Despite recent insights, tumor cell-intrinsic and -extrinsic Axl functions are poorly understood in hepatocellular carcinoma (HCC). Thus, we analyzed the cell-specific aspects of Axl in liver cancer cells and in the tumor microenvironment. We show that tumor-intrinsic Axl expression decreased the survival of mice and elevated the number of pulmonary metastases in a model of resection-based tumor recurrence. Axl expression increased the invasion of hepatospheres by the activation of Akt signaling and a partial epithelial-to-mesenchymal transition (EMT). However, the liver tumor burden of Axl^+/+^ mice induced by diethylnitrosamine plus carbon tetrachloride was reduced compared to systemic Axl^−/−^ mice. Tumors of Axl^+/+^ mice were highly infiltrated with cytotoxic cells, suggesting a key immune-modulatory role of Axl. Interestingly, hepatocyte-specific Axl deficiency did not alter T cell infiltration, indicating that these changes are independent of tumor cell-intrinsic Axl. In this context, we observed an upregulation of multiple chemokines in Axl^+/+^ compared to Axl^−/−^ tumors, correlating with HCC patient data. In line with this, Axl is associated with a cytotoxic immune signature in HCC patients. Together these data show that tumor-intrinsic Axl expression fosters progression, while tumor-extrinsic Axl expression shapes an inflammatory microenvironment.

## 1. Introduction

Gas6/Axl signaling has emerged as a driver of hepatocellular carcinoma (HCC) progression, impacting several cellular processes, including the epithelial-to-mesenchymal transition (EMT) and metastasis [1]. Moreover, immune-modulatory functions of Axl affect the tumor microenvironment (TME), which unveils additional aspects for developing therapeutic strategies in cancer [2].

HCC is the major subtype of liver cancer and represents a highly lethal malignancy with rising incidence [3]. The majority of HCC cases develop in the background of cirrhosis caused by chronic liver inflammation. Chronic infection with hepatitis B or C virus, non-alcoholic steatohepatitis (NASH) and permanent alcohol abuse are predominant causes for developing cirrhosis and HCC [4]. Immunotherapy in combination with anti-angiogenic treatment has replaced first-line therapy with tyrosine kinase inhibitors. In this context, the classification of HCC into inflamed and non-inflamed subclasses, the latter containing the majority of HCC patients, has been proposed. Non-inflamed tumors indicate low expression of immune checkpoints together with minor or even no immune cell infiltration, resulting in less suitability for immunotherapy [5,6].

Most HCC patients are diagnosed at advanced stages of disease characterized by the vascular invasion of cancer cells and their dissemination to intra- or extrahepatic sites [3]. The heterogeneity of HCC cells is associated with changes in epithelial cell plasticity, which most commonly includes EMT of neoplastic hepatocytes or hepatic progenitor cells [7]. A large panel of EMT regulators, acting at the transcriptional and post-transcriptional level, causes the dissociation of HCC cells from epithelial organization, allowing for individual cell movement and local cell invasion into the surrounding tissue [8]. The reprogramming from the epithelial to mesenchymal state encompasses the impaired expression of the cell adhesion constituent E-cadherin and the upregulation of EMT transcription factors together with cytoskeletal modulators [7]. Recent advances describe EMT as an intermediate continuum displaying quasi-mesenchymal phenotypes termed the partial EMT, which are essentially involved in the metastatic process. Cancer cells exhibiting partial EMT assemble an immunosuppressive tumor microenvironment (TME), which is refractory to immune checkpoint blocking therapy [9,10]. Yet the role of the partial EMT in HCC and its link to Axl remains to be elucidated.

The receptor tyrosine kinase (RTK) Axl belongs—together with MerTK and Tyro3—to the family of TAM receptors and has been identified as a major driver of cancer progression. Upon binding to its ligand Gas6 and homodimerization, phosphorylated Axl activates numerous signaling pathways, including MAPK, Akt, Src and Jak/Stat. In cancer, the upregulation of Axl is frequently associated with tumor cell survival, EMT, angiogenesis and drug resistance [2]. Indeed, the impact of Axl in promoting EMT, resistance to targeted therapy and the induction of a pro-tumorigenic TME is comprehensively described in breast and lung cancer [11,12]. In HCC, elevated Axl expression is found in a large proportion of patients, which correlates with vascular invasion and a poor survival outcome. In line with this, mesenchymal-like HCC cells express high levels of Axl and display increased migratory characteristics compared to epithelial HCC cells, highlighting the role of Axl in cellular plasticity [1,13,14,15]. Yet the role of tumor cell-intrinsic Axl expression in modulating neoplastic epithelial liver cancer cells and its mechanistic contribution to HCC dissemination is still poorly understood. It is noteworthy that the multi-tyrosine kinase inhibitor Cabozantinib, which targets Axl among other RTKs, is used as a third-line treatment option for patients with advanced HCC, supporting the potential clinical relevance of Axl in this disease [16].

Besides its expression in tumor cells, Axl has established roles in a variety of stromal cells. Physiologically, Axl is expressed on endothelial cells, platelets, fibroblasts and immune cells, including dendritic cells (DCs), macrophages and natural killer (NK) cells [17]. Axl is mainly involved in maintaining immune homeostasis by suppressing TLR signaling and reducing pro-inflammatory cytokines, which activate DCs, macrophages and NK cells [18]. TAM receptor triple-knockout mice suffer from severe immunological adverse events due to the inability to phagocytose apoptotic cells resulting in the elevated release of self-antigens [19]. Furthermore, Axl plays a crucial role in the maturation of NK cells, emphasizing its impact in immune cell function [20]. Besides its immune-modulatory roles, Gas6/Axl signaling contributes to endothelial cell function by regulating vascular integrity [21]. Moreover, Axl expression promotes the stabilization of platelets and therefore thrombus formation [22]. In liver fibrosis, Axl is required for the activation of hepatic stellate cells and trans-differentiation to myofibroblasts, thereby driving disease progression [23]. These findings underline the versatile function of Axl in stromal cells and emphasize the relevance of investigating the impact of tumor cell-extrinsic Axl expression.

Despite evidence that Axl has a disease-promoting role in HCC cells and fibrosis, it is unclear how tumor cell-intrinsic and -extrinsic Axl expression shapes the development and progression of HCC. In this study, we show, on the one hand, that tumor cell-intrinsic Axl expression induces Akt signaling and promotes partial EMT, cell invasion and metastatic colonization of lungs. On the other hand, we show that tumor cell-extrinsic Axl expression modulates the TME towards an immune-activated, “hot” tumor in mice and HCC patients, suggesting a novel role of Axl in HCC.

## 2. Results

### 2.1. Axl Augments Invasive Behavior of HCC Cells

We recently showed that Axl expression correlates with vascular invasion in HCC patients [1]. To assess the phenotypical consequences of Axl expression in neoplastic hepatocytes, we used murine MR and human Hep3B cells and generated cells with transgenic Axl expression. Murine Axl-deficient MR hepatocytes harbor oncogenic H-Ras to induce malignant transformation including tumor formation in vivo [24]. The re-expression of Axl in MR and Hep3B, resulting in MR-Axl and Hep3B-Axl cells, respectively, was confirmed by Western blotting (Figure 1A,B). Axl expression did not affect the proliferation of the MR and Hep3B cells in vitro (Figure 1C,D). In line with this, neither the stimulation of the cells with Gas6 nor the inhibition of Gas6/Axl signaling using an Axl decoy receptor modulated proliferation (Appendix A). In addition, tumor growth in immunocompromised SCID mice was not dependent on Axl expression as indicated by the tumor volume and absolute tumor weight (Figure 1E,F and Appendix A). In line with this, the proliferation of cancer cells in MR- and MR-Axl-derived tumors was not affected as determined by the immunohistochemistry (Appendix A). However, Axl expression caused a 2.5-fold and 9.2-fold increase in the invasion of MR-Axl- and Hep3B-Axl-derived spheroids compared to MR and Hep3B-GFP spheroids when stimulated with Gas6 (Figure 1G,H). Together, these data show that Axl does not promote proliferation but induces invasiveness of murine and human malignant hepatocytes.

### 2.2. Axl Promotes Metastasis In Vivo

To investigate the impact of Axl on the metastatic potential of MR cells, we established a resection-based tumor recurrence model. Thus, we generated subcutaneous MR and MR-Axl tumors in SCID mice and resected them after they reached 500 mm^3^. Subsequently, a Kaplan–Meier analysis was performed to examine whether Axl has an impact on metastatic colonization. Notably, MR-Axl tumors significantly reduced the survival of mice compared to those harboring Axl-deficient tumors (Figure 1I). The survival of mice with MR-Axl tumors was slightly improved by the treatment with the Axl decoy receptor (Figure 1I), which failed to show changes in the survival of mice with MR tumors (Appendix A). Furthermore, intravenous injection of MR and MR-Axl cells showed that the tumor load of lungs in mice was not significantly affected by Axl expression as indicated by the lung/body weight ratio and tumor/lung area ratio (Figure 1J,K). Yet the number of MR-Axl tumor nodules in the lung was significantly increased, suggesting that Axl supports the extravasation and metastatic colonization of lungs (Figure 1L). From these data, we concluded that tumor cell-intrinsic Gas6/Axl signaling does not control proliferation but augments invasiveness and the metastatic potential of malignant hepatocytes in vitro and in vivo.

### 2.3. Axl Is Associated with Partial EMT

To investigate whether the Axl-dependent, invasive phenotype of MR-Axl and Hep3B-Axl correlates with changes in epithelial plasticity, we analyzed the expression of EMT constituents by qPCR. We observed that Vimentin, Snai1 and Snai2 were upregulated in MR-Axl cells compared to MR cells (Figure 2A). Similarly, the expression of Vimentin was highly increased by exogenous Axl expression in the human liver cancer cell lines Hep3B and Huh7, while the expression of E-Cadherin was downregulated, emphasizing the role of Axl in the EMT (Figure 2B,C). Moreover, the expression of Vimentin and Snai1 in HCC patients correlated with Axl expression (Figure 2D,E), and additionally, elevated E-cadherin expression was accompanied by increased survival of HCC patients (Figure 2F). The expression of E-cadherin protein was significantly downregulated in MR-Axl spheroids, suggesting that Axl promotes the mesenchymal phenotype of malignant hepatocytes (Figure 2G and Appendix A). Moreover, we examined the expression of E-cadherin and Vimentin in subcutaneous tumors derived from MR and MR-Axl cells, which did not exhibit significant changes (Appendix A). Together, these data show that Axl is associated with the expression of EMT-associated genes in human and murine liver cancer cell lines, suggesting a partial EMT phenotype.

### 2.4. Gas6/Axl/Akt Signaling Mediates Invasion of Liver Cancer Cells

Next, we investigated which Gas6/Axl-mediated pathways promote the invasive phenotype of liver cancer cells. Thus, we performed a phospho-proteomic analysis of MR and MR-Axl cells in the presence and absence of Gas6. A principal component analysis showed sample-dependent clustering (Figure 3A). Protein clustering based on the pattern of phosphorylation across the conditions and samples revealed four clusters: phospho-sites that are upregulated dependent on Axl expression (cluster 1) or Gas6 expression (cluster 2) and phospho-sites that are downregulated dependent on Axl expression (cluster 3) or Gas6 expression (cluster 4) (Appendix A). A functional enrichment analysis showed the enrichment of protein sets associated with cellular polarity and motility (KEGG Pathways: Regulation of the Actin Cytoskeleton; MSigDB Hallmark: Tight Junctions, Apical Junctions and Adherens Junctions) (Appendix A), depending on the changes in Axl expression and Gas6 activation. Furthermore, we analyzed the phospho-proteomic data to estimate the activity of kinases based on kinase–substrate interactions using PhosR [25]. A kinase perturbation analysis indicated the mammalian target of rapamycin (mTOR) as the one with the highest upregulated activity (*p =* 0.06; one-sided Pearson’s method; [26]) in Gas6-stimulated MR-Axl cells compared to Gas6-stimulated MR cells (Figure 3B). Accordingly, we analyzed the substrates of mTOR and found that the phosphorylation of Akt (Ser473) was upregulated in cluster 1 and highly increased in Gas6-stimulated MR-Axl and Hep3B-Axl cells, suggesting that Akt is regulated by Axl. To examine whether Akt signaling is regulated by Gas6/Axl signaling, we treated cells with Gas6 and concomitantly inhibited Akt using MK2206, resulting in the reduced phosphorylation of Akt in MR-Axl and Hep3B-Axl cells (Figure 3C,D). In agreement with these data, the inhibition of Akt decreased the invasive phenotype of Gas6-stimulated MR-Axl and Hep3B-Axl tumor spheroids, indicating that Gas6/Axl/Akt signaling mediates the invasion of liver cancer cells (Figure 3E,F).

### 2.5. Tumor Cell-Extrinsic Axl Signaling Promotes Proliferation of Neoplastic Hepatocytes

We further focused on the cell-specific role of Axl in endogenous mouse models of liver tumorigenesis. We induced liver tumor formation and inflammation in mice harboring a hepatocyte-specific deletion of Axl (Axl^Δhep^) and mice with a systemic Axl knockout (Axl^−/−^) by administering DEN+CCl_4_. Axl^fl/fl^ and Axl wild-type (Axl^+/+^) mice were used as the respective controls (Figure 4A,B). First, we confirmed the published data demonstrating that Axl expression increases hepatic fibrosis compared to Axl^−/−^ mice but is not altered between Axl^fl/fl^ and Axl^Δhep^ mice (Appendix A). The proliferation of tumor cells as indicated by the relative abundance of BrdU+ cells was not affected between Axl^fl/fl^ and Axl^Δhep^ mice but was significantly increased in the Axl^−/−^ mice, indicating that tumor cell-extrinsic Axl expression impacts liver tumor growth (Figure 4C,D). In accordance with this, no difference in liver tumor burden was observed between Axl^fl/fl^ and Axl^Δhep^ mice (Appendix A). We detected a slight, but not significant, increase in tumor burden in Axl^−/−^ mice (*p =* 0.0822), while the absolute body weights of mice were not altered (Appendix A). Moreover, liver tumor mass (tumor area/liver area and tumor number/section) of Axl^fl/fl^, Axl^Δhep^, Axl^+/+^ and Axl^−/−^ mice was not changed between the groups, suggesting that Axl expression does not affect primary tumor burden (Appendix A). Interestingly, the analysis of Axl expression in DEN+CCl_4_-treated livers of the Axl^fl/fl^, Axl^Δhep^ and Axl^+/+^ mice revealed that Axl is expressed in stromal cells rather than in neoplastic hepatocytes, explaining the lack of a tumor phenotype by comparing Axl^fl/fl^ and Axl^Δhep^ mice (Appendix A). Together, these data show that tumor cell-extrinsic Axl expression decreases the proliferation of malignant hepatocytes.

### 2.6. Axl Stimulates Infiltration of Cytotoxic Immune Cells

Based on the regulatory role of Axl on immune cells and on our observation that extrinsic Axl expression reduces tumor cell proliferation, we investigated the impact of Axl on the immune microenvironment of DEN+CCl_4_-induced liver tumors. Notably, we found that cytotoxic CD8+ T cells highly infiltrated Axl^+/+^ tumors compared to Axl^−/−^ tumors (Figure 5A,B), whereas no changes were detected in the myeloid immune cell compartment (Appendix A). We further found that Axl^+/+^ tumors were significantly more infiltrated with Granzyme B+ cells, showing cytotoxic activity in those tumors (Appendix A). Yet the number of PD-L1+ cells did not vary between the Axl^+/+^ and Axl^−/−^ tumors, suggesting that Axl does not affect tumor immune escape via the PD-1/PD-L1 axis (Appendix A and Figure 5A,C). Notably, the livers of the untreated Axl^+/+^ and Axl^−/−^ mice did not display any differences in the relative abundance of CD8+ T cells, CD4+ T cells and FoxP3+ regulatory CD4+ T cells, indicating that Axl does not alter the healthy liver environment (Appendix A). Moreover, we neither observed changes in the CD8+ T cell infiltration nor myeloid cell subsets in Axl^fl/fl^ and Axl^Δhep^ mice (Figure 5C,D and Appendix A), suggesting that the tumor cell-extrinsic expression of Axl is responsible for the elevated number of CD8+ T cells in Axl^+/+^ tumors. In summary, these data show that tumor cell-extrinsic Axl expression impacts the infiltration of CD8+ and Granzyme B+ cells to liver tumors.

### 2.7. Ccl9/Ccl15 Negatively Correlates with Axl Expression

In a further step, we aimed to obtain comprehensive insights into the inflammatory milieu of DEN+CCl_4_ tumors by cytokine profiling of tissue lysates of Axl^+/+^ and Axl^−/−^ mice. We detected six significantly upregulated (Cxcl2, Cxcl4, P-Selectin, Vcam1, Vegf-a and sTnfR I) and one downregulated (Ccl9) cytokine in Axl^+/+^ tumors (Figure 5E and Appendix A). Ccl2 (*p =* 0.0886) and Cxcl5 (*p =* 0.0532), which has been associated with Axl expression, were borderline significantly upregulated in Axl^+/+^ tumors (Figure 5E) [27]. Notably, the expression of these deregulated cytokines was not altered on the transcript level in Axl^fl/fl^ and Axl^Δhep^ tumors (Figure 5F). Accordingly, we found that the expression of Ccl15, the human homologue to Ccl9, negatively correlated with Axl expression in HCC patient tissue but not in normal liver tissue (Figure 5G,H). In line with this, *CCL2*, *CXCL2*, *CXCL4*, *P-Selectin* and *VCAM1*, but not *VEGF-A*, positively correlated with *Axl* expression in the HCC patient tissue but to a lesser extent in normal liver tissue (Figure 5I,J and Appendix A). In conclusion, we found that several chemokines were upregulated in tumors of the Axl^+/+^ mice, which also positively correlated with the expression of Axl in human HCCs, suggesting the augmented activation of immune cells depending on Axl expression in tumors.

### 2.8. Tumor Cell-Extrinsic Axl Expression Correlates with the Infiltration of CD8+ and Granzyme B+ Cell in HCC Patients

Given the correlation of Axl expression and elevated CD8+ and Granzyme B+ cells in DEN+CCl_4_-induced liver tumors, we investigated whether a comparable phenotype could be observed in HCC patients. First, we observed that the expression of immune cytotoxic effector molecules such as *Granzyme A*, *Granzyme B* and *Perforin-1* positively correlated with *Axl* expression in HCC patients (Figure 6A–C). In accordance with this, expression of *Axl* and *Granzyme A*, *Granzyme B* or *Perforin-1* was also associated with an increased survival probability of HCC patients (Figure 6D–F). To investigate whether the infiltration of CD8+ and Granzyme B+ cells correlates with Axl expression in tumor cells, we analyzed tissue samples of 40 HCC patients for the infiltration of CD8+ T cells (CD45^+^ CD8^+^), Granzyme B+ cells (CD45^+^ Granzyme B^+^) and Axl expression in tumor cells (Axl^+^ CD31^-^ αSMA^−^ CD45^−^) or immune cells (Axl^+^ CD45^+^) by multiplexed immunohistochemistry (Figure 6G). Interestingly, we did not observe a correlation between Axl expression on the tumor cells (Appendix A). However, we observed that patients with a high infiltration of CD45^+^ CD8^+^ T and CD45^+^ Granzyme B^+^ cells harbored tumors with an increased infiltration of CD45^+^ Axl^+^ cells (Figure 6H,I). Interestingly, high levels of Axl^+^ CD45^+^, CD45^+^ CD8^+^ and Axl^+^ CD31^−^ alpha-smooth muscle actin (αSMA)^−^ CD45^−^ (tumor cells) did not correlate with survival probability (Figure 6J and Appendix A) but increased the infiltration of CD45^+^ Granzyme B^+^ associated with an increased survival probability (Figure 6K). Together, these data suggest that Axl expression on CD45+ cells, rather than on tumor cells, correlates with the increased infiltration of CD8+ and Granzyme B+ in HCC patients.

## 3. Discussion

We examined the tumor cell-intrinsic and -extrinsic role of Axl in mouse models of liver tumorigenesis and HCC patients. We found that tumor cell-intrinsic Axl expression does not impact tumor cell proliferation but promotes cancer cell invasion, leading to increased pulmonary metastasis and reduced survival in mice. The Axl-dependent invasive phenotype is associated with the expression of mesenchymal genes and downregulation of E-cadherin, suggesting that Axl promotes a partial EMT. A phospho-proteomic analysis revealed that Akt signaling was upregulated in Axl-expressing tumor cells and that pharmacological intervention with Akt signaling reduced their invasive abilities. In an endogenous mouse model of liver tumorigenesis, tumor cell-extrinsic expression decreased tumor proliferation. Immune cell profiling of Axl^+/+^ and Axl^−/−^ tumors showed increased CD8+ T cell infiltration in Axl-proficient tumors, which was accompanied by an elevated infiltration of Granzyme B+ cells. In accordance with this, we found that Axl expression on CD45+ cells positively correlated with the infiltration of CD45+ CD8+ and CD45+ Granzyme B+ cells in human HCC tissue.

In the context of tumor cell-intrinsic expression, Axl has an established role in the fields of EMT and drug resistance. In renal clear cell carcinoma, Axl is activated by HIF-1 and HIF-2, leading to the activation of the met proto-oncogene through Src signaling, which maximizes cancer cell invasion [28]. In head and neck cancer, the mutated Ras oncogene stabilizes YAP1 activity to promote Axl-mediated cell invasion in vitro and metastasis in vivo [29]. In HCC, Axl induces the expression of PRAME through MAPK signaling, which causes the dedifferentiation of cancer cells, a loss of liver function and elevated cancer cell invasion [30]. In line with these data, we showed that tumor cell-intrinsic Axl expression augments cancer cell invasion but does not affect the proliferation of tumor cells in vitro and in vivo.

Furthermore, we observed that the invasive phenotype, directed by exogenous Axl expression, promoted the expression of EMT-associated genes, suggesting the acquisition of a partial EMT phenotype. In accordance with these data, Axl expression is frequently upregulated in mesenchymal-like HCC cells and enhances cancer cell motility [1,13,14,15]. However, recent studies focused on the role of Axl in HCC cells, which were already EMT-transformed and harbor the disadvantage of not forming tumors in transplant-based animal models. In this study, we demonstrate that Gas6/Axl signaling supports a partial EMT in MR-Axl cells in the background of oncogenic H-Ras and, importantly, strongly reduces the survival of mice in a model of resection-based tumor recurrence in vivo together with increased abilities to form metastatic colonies in lungs. These models indicate that Axl drives early stages of the metastatic cascade, e.g., EMT and intravasation, which promote metastatic colonization.

Notably, EMT is a dynamic process where cells are rarely fully epithelial or mesenchymal but rather express both epithelial and mesenchymal markers. Therefore, cancer cells expressing both epithelial and mesenchymal markers are partially transformed, resulting in a hybrid state of EMT. This so-called partial EMT phenotype has been associated with increased tumorigenicity and invasiveness and is found to be more aggressive than mesenchymal-like tumor cells [31,32]. Axl expression has been linked to a partial EMT in breast cancer, but its role in mediating a partial EMT in HCC is an unresolved issue [33]. Thus, we hypothesize that Axl promotes changes in epithelial cell plasticity and that MR-Axl cells resemble a partial EMT phenotype that supports invasiveness as they express less E-cadherin in vitro while upregulating the expression of EMT transcription factors [9]. In accordance with this, we found a slight downregulation of E-cadherin-expressing cells in MR-Axl tumors, supporting the hypothesis that the cells are partially EMT-transformed.

A phospho-proteomic analysis revealed that Axl induces the phosphorylation of Akt at Ser473 and inhibiting Akt signaling reduced the invasiveness of MR-Axl tumor spheroids. A previous study showed that Axl-dependent Akt phosphorylation increases cancer cell invasion, highlighting the role of Akt signaling in mediating invasion [34]. Additionally, Axl/Akt signaling enhanced the metastatic potential of a murine mouse model of ovarian cancer without affecting tumor cell proliferation, which is in line with our findings [35]. Moreover, the PhosR analysis revealed the activation of mTOR in Gas6-stimulated MR-Axl cells. mTOR forms two complexes, mTORc1 and mTORc2, which have distinct binding partners, substrates and cellular functions [36]. We found that Akt is phosphorylated at Ser473, suggesting that mTORc2 is activated upon Gas6/Axl signaling and that Gas6/Axl/mTOR/Akt signaling induces the invasive phenotype of MR-A cells. However, further studies are needed to examine the involvement of mTOR in Gas6/Axl signaling.

Next, we addressed the role of tumor cell-intrinsic and -extrinsic Axl expression in liver tumorigenesis by treating mice with DEN and CCl_4_. By employing systemic Axl^−/−^ and Axl^+/+^ mice, we investigated how stromal Axl expression impacts liver tumor formation. Notably, we observed that the liver/body weight ratio was slightly increased in Axl^−/−^ mice. However, as Axl^−/−^ mice exhibit less fibrosis than Axl^+/+^ mice, the difference in the liver/body weight ratio might stem from reduced parenchymal shrinkage. This could explain the similar tumor/liver areas in Axl^+/+^ versus Axl^−/−^ mice. Interestingly, we found that systemic Axl deficiency increased tumor cell proliferation. When examining the TME, we found that CD8+ and Granzyme B+ cells were highly infiltrated in Axl^+/+^ tumors compared to Axl^−/−^ tumors. First, we hypothesized that this phenotype stems from the expression of immune checkpoint molecules because tumor cell-intrinsic Axl expression is associated with decreased expression of major histocompatibility complex class 1 (MHC-I) and increased PD-L1 expression, supporting immune escape and therefore tumor progression [37,38]. In addition, few studies have shown that Axl inhibition induces the expression of PD-L1 and thus increases efficacy of PD-L1 targeting [39,40]. In this context, we did not detect changes in PD-L1 expression. Hence, we hypothesized that liver tumors from Axl^+/+^ and Axl^−/−^ mice might already escape immune surveillance. However, this hypothesis needs further testing and should be evaluated in earlier timepoints of liver tumorigenesis. Furthermore, we assumed that the number of DCs could be altered in Axl^−/−^ mice, resulting in decreased CD8+ T cell infiltration. However, we did not observe any differences in DC infiltration. Intriguingly, a study showed that the maturation of DCs is diminished in Axl^−/−^ mice resulting in limited T cell priming [41]. This raises the question whether DCs in Axl^−/−^ mice are capable of sufficient antigen presentation and therefore T cell activation. Interestingly, Gas6/Axl signaling has been identified as a key regulator of NK cell maturation and functionality [20]. Based on these data, we speculate that the decreased number of Granzyme B+ cells in Axl^−/−^ tumors could stem from diminished NK cell development and activation.

Moreover, we observed that several chemokines/cytokines were upregulated in Axl^+/+^ tumors, while the chemokine Ccl9 was downregulated. In accordance with this, we found that the expression of these chemokines and cytokines correlated with Axl in HCC patients. Based on these findings, we suppose that Axl^+/+^ tumors were highly infiltrated with activated immune cells, suggesting that Axl^+/+^ mice harbor immune “hot” tumors that could be targeted for therapies. Interestingly, Ccl15, the human homologue to Ccl9, has been identified as a biomarker of HCC and correlates with the augmented migratory and invasive potential of HCC cells [42]. Additionally, a study showed that Ccl15 expression in HCC patients correlates with a poor clinical outcome, increased tumor cell invasion and recruitment of immune-suppressive CD14+ monocytes [43]. It is noteworthy that the link between Axl deficiency and Ccl9/15 has yet to be elucidated but highlights our findings that Ccl9/15 expression correlates with increased tumor burden and a pro-tumorigenic microenvironment.

In line with our endogenous mouse model, a study of inflammation-associated colon cancer showed that Axl/MerTK deficiency led to increased tumor burden. Interestingly, this phenotype was linked to an augmented production of pro-inflammatory cytokines and elevated numbers of apoptotic neutrophils, generating a tumor-promoting environment [44]. Therefore, we hypothesize that Axl signaling is involved in decreasing tumor-promoting inflammation in normal tissue prior to malignant transformation, which could explain the reduced tumor formation in Axl^+/+^ mice. Further studies are needed to determine the timepoint when the liver environment switches from anti- to pro-inflammatory and how Axl shapes the immune microenvironment and therefore liver cancer development.

In contrast to our transplantation-based mouse model and previous findings in HCC patients, we found that Axl expression on tumor cells did not correlate with a decreased survival outcome in our HCC patient cohort [1]. We detected a positive correlation between the infiltration of Axl+ CD45+ and CD45+ CD8+ and CD45+ Granzyme B+ cells in HCC tissue. However, we observed that only patients with a high infiltration of CD45+ Granzyme B+ had a survival benefit. Due to the experimental limitations, we could not analyze the specific cell type of Axl expressing CD45+ cells and the activation or exhaustion status of CD45+ CD8+ T cells. Especially, the activation status of CD8+ T cells in HCC is of great interest, because recent studies suggest that a population of CD8+ T cells in NASH-HCC patients fosters tumor progression [45]. Importantly, an elevated number of HCC specimens allowing for the stratification of patients is needed to further validate and investigate our findings.

Unfortunately, we could not detect Axl expression in the hepatocytes of Axl^+/+^ and Axl^fl/fl^ mice upon liver tumor formation. As Axl gets upregulated during the EMT, we hypothesize that the expression is not upregulated in Axl^+/+^ and Axl^fl/fl^ tumors because neoplastic hepatocytes in DEN-induced tumors are well differentiated and neither undergo EMT nor metastasize. Therefore, overexpression of Axl in an endogenous mouse model would be necessary to better understand the role of Axl in liver tumorigenesis. For example, the introduction of Axl together with the Sleeping Beauty transposon system into the liver by hydrodynamic tail vein injection could be combined with DEN-dependent hepatocarcinogenesis [46,47]. Moreover, several studies have shown that the TME is highly impacted by the disease background in HCC [45,48]. However, the DEN+ CCl_4_ model cannot be assigned to a specific etiology of HCC, thus limiting our findings. Therefore, utilizing disease-specific models of HCC, for example, a diet- and chemical-induced model of hepatocarcinogenesis, is required to validate our findings [49]. In addition, further efforts using conditional knockout mice and in vitro co-culture experiments are needed to determine the specific cell types mediating the immune phenotype of Axl^+/+^ tumors.

Based on the dichotomic role of Axl, i.e., that tumor cell-intrinsic expression fosters tumor progression, while tumor cell-extrinsic Axl expression generates an immunogenic tumor, unspecific targeting Axl may have unwanted adverse effects. To circumvent this problem, generating bi- or tri-specific antibodies targeting Axl and tumor cell-specific antigens could induce more specific targeting of tumor cell-intrinsic Axl [50,51,52]. Even though Axl has promising potential in cancer therapy, the complex interplay between tumor cell-intrinsic and -extrinsic Axl expression highlights the need for further research to investigate its specific mechanisms and optimize future treatment strategies.

In this study, we show that tumor cell-intrinsic Axl expression promotes the invasion of liver cancer cells by Akt signaling and induces a partial EMT leading to metastatic colonization. In this respect, targeting Axl could be used for treating advanced HCC patients and thus harnessing tumor progression. Moreover, we provide the first evidence in HCC that tumor cell-extrinsic Axl expression modulates the tumor immune microenvironment by recruiting CD8+ T cells and Granzyme B+ cells. Close to the findings in mouse models, we observed that Axl expression on CD45+ cells correlated with the elevated infiltration of CD8+ and Granzyme B+ cells in HCC patients (Appendix A). Hence, tumor cell-extrinsic Axl expression could predict the therapy response in HCC patients, as they display infiltrated tumors of the “inflamed” subclass that respond to the currently used immune check point blockade. Together, this study demonstrates the versatile functions of Axl dependent on the cellular context, which are relevant for developing novel strategies to combat HCC.

## 4. Material and Methods

### 4.1. Cell Culture

Hepatocytes were isolated from livers of p19^ARF−/−^/Axl^−/−^ mice after perfusion as described [53]. Notably, p19^ARF^ deficiency allows for the immortalization of murine hepatocytes without losing genetic stability. The cells were further virally transduced with a vector expressing oncogenic H-Ras to generate neoplastic MR cells lacking p19^ARF^ and Axl [24]. To investigate the role of Axl, MR cells were transduced with a vector (pWPI; Addgene, Cambridge, MA, USA) expressing wild-type Axl and GFP to generate MR-Axl cells. MR and MR-Axl cells were cultured on collagen-coated dishes. MR, MR-Axl, Huh7 and Huh7-Axl were cultured in RPMI-1640 (Life Technologies; Carlsbad, CA, USA) supplemented with 10% FCS (Life Technologies; Carlsbad, CA, USA) and 100 U/mL Penicillin/Streptomycin (Life Technologies; Carlsbad, CA, USA) at 37 °C and 5% CO_2_. Hep3B-GFP and Hep3B-Axl were cultured in EMEM (Life Technologies; Carlsbad, CA, USA) supplemented with 10% FCS and 100 U/mL Penicillin/Streptomycin at 37 °C and 5% CO_2_. The Huh7 and Hep3B cell lines were purchased from ATCC. The identity of the cell lines was confirmed by a short tandem repeat analysis. All the cells were routinely screened for the absence of mycoplasma.

### 4.2. Patient Samples

Patients were enrolled at the General Hospital of Vienna/Medical University of Vienna and the hospital Klinik Landstrasse (Austria). The inclusion criteria encompassed patients aged between 18 and 90 years undergoing surgical resection for primary or secondary liver malignancies, with prerequisites to adhere to the study protocol and signing an informed consent. Based on the guidelines of the European Association for the Study of the Liver, HCC was diagnosed by histology or dynamic imaging, such as computed tomography or magnetic resonance imaging [54]. In accordance with the Declaration of Helsinki from 1964 including the current revisions, this study was approved by the ethics committee of the Medical University of Vienna (1186/2018). All patients signed an informed consent form prior to inclusion in this study. For the Kaplan–Meier analysis, the patients were separated into “high” and “low” CD45+ CD8+, CD45+ Granzyme B+, Axl+ CD45+ and Axl+ CD31- αSMA- CD45- tumor infiltration.

### 4.3. In Silico Analysis

The gene expression data from normal adjacent liver and primary HCC tissue were retrieved from The Cancer Genome Atlas (TCGA) Liver Cancer cohort (LIHC), which was downloaded from the open-source Xenabrowser [55]. The Pearson correlation co-efficient was determined using GraphPad Prism 5 (GraphPad Software Inc., San Diego, CA, USA). The Kaplan–Meier plots of the HCC patients based on mRNA expression were generated using the Kaplan–Meier Plotter “https://kmplot.com/analysis(accessed on 5 May 2022)” [56].

### 4.4. Proliferation Kinetics

In total, 2.5 × 10^4^ of each of the MR and MR-Axl cells were seeded in triplicate into 12-well plates to perform cumulative proliferation kinetics as described [24]. The MR and MR-Axl cells were treated either with the Gas6 (500 ng/mL), Axl decoy (50 µg/mL), combination or vehicle (phosphate-buffered saline, PBS). In total, 2 × 10^4^ cells of the Hep3B-GFP and Hep3B-Axl cells were seeded in triplicate into 12-well plates. The cell numbers were determined every second to third day for 7 days using the CASY cell counter (Schärfe Systems, Reutlingen, Germany).

### 4.5. Invasion Assay

In total, 1 × 10^3^ MR, MR-Axl, Hep3B-GFP or Hep3B-Axl cells were seeded into U-bottom-shaped ultra-low attachment plates (Greiner Bio-One, Kremsmünster, Austria) and incubated for 72 h to form spheres. After incubation, the spheroids were embedded in a 1.5 mg/mL rat-tail collagen I supplemented with 0.125% sodium bicarbonate, 15 mM HEPES and 2% FCS as described [57]. Gas6 (500 ng/mL), MK2206 (1.25 µM) or the respective vehicle were added into the collagen matrix and media prior to incubation for 24 h. The images of the spheres were taken by the TiEclipse phase contrast microscopy (Nikon, Tokyo, Japan) and analyzed using ImageJ version 1.38 (ImageJ, Bethesda, MD, USA). The invaded area was calculated by subtracting the core area of the spheroid from the total spheroid area.

### 4.6. Quantitative PCR (qPCR)

RNA was isolated from cells using the Monarch total RNA isolation kit (New England Biolabs, Ipswich, MA, USA). The RNA was transcribed into cDNA using the cDNA iScript^TM^ kit (BioRad, Hercules, CA, USA). For the quantification of mRNA expression, the LUNA qPCR Master Mix (New England Biolabs, Ipswich, MA, USA) and the BioRad CFX Connect Real-Time PCR system (BioRad, Hercules, CA, USA) were used. The initial denaturation was performed at 95 °C for 1 min, followed by 40 cycles of denaturation at 95 °C for 15 s, annealing at 60 °C for 30 s and extension at 60 °C for 1 min. The primers specific for individual targets are listed in Appendix A. *28S* was used as the housekeeping gene. The expression of the transcripts was calculated using the CFX Maestro Software version 4.0.2325.0418 (BioRad, Hercules, CA, USA). The 2^–∆∆Ct^ method was used for the quantification of the RT qPCR [58].

### 4.7. Immunofluorescence Analysis of Spheroids

MR and MR-Axl spheroids were each collected in PBS and centrifuged onto glass slides by cytospin (Epredia^TM^; Kalamazoo, MI; USA). Afterwards, the spheroids were fixed in cold methanol/acetone (1:1) for 20 min. The cells were washed with PBS and blocked for 1 h with mouse IgG (M.O.M Blocking reagent; Agilent; Santa Clara, CA, USA) and 30 min with 5% goat serum in PBS. After blocking, the spheroids were washed and incubated with anti-E-cadherin (1:50; mouse; BD Biosciences; #610182) and anti-actin (1:50; rabbit; Abcam, Waltham, MA, USA; #A2066) antibodies for 1 h at room temperature. Next, the spheroids were washed and incubated with goat anti-rabbit Alexa Fluor488 (1:200; Invitrogen; #A-11008) and goat anti-mouse Alexa Fluor647 (1:200; Invitrogen; #A-11001) in 5% goat serum/PBS for 1 h at room temperature. Finally, the spheroids were incubated with Hoechst dye (5 µg/mL) for 5 min and washed. The spheroids were mounted with mounting medium (ProLong^TM^ Glass Antifade, Invitrogen^TM^; Waltham, MA, USA) and dried overnight at 4 °C before imaging with the IXplore Spinning Disk Confocal system (Olympus; Tokyo; Japan). The intensities of E-cadherin and actin expression were analyzed using the cellSens software version 3.2 (Olympus; Tokyo; Japan). E-cadherin expression was normalized to actin expression.

### 4.8. Phospho-Proteomics by Mass Spectrometry (MS)

To identify the Gas6/Axl-regulated phosphorylated events, MR and MR-Axl cells were stimulated with Gas6 (500 ng/mL) or the vehicle (PBS). The cells were washed three times with PBS and collected in a 1.5 mL tube, centrifuged and snap frozen. The cell pellets were analyzed at the Mass Spectrometry Facility of the Max Perutz Labs (Vienna, Austria). After lysis of the cells, an analysis of the proteins was performed on an UltiMate 3000 Dual Liquid chromatography (LC) nano-HPLC System (Dionex, Thermo Fisher Scientific), containing both a trapping column for the peptide concentration (PepMap C18, 5 × 0.3 mm, 5 μm particle size) and an analytical column (PepMap C18, 500 × 0.075 μm, 2 μm particle size; Thermo Fisher Scientific, Waltham, MA, USA), coupled to an Orbitrap Exploris 480 mass spectrometer (Thermo Fisher) via a FAIMS Pro ion source (Thermo Fisher). The details of the sample preparation and LC/MS analysis are described in the Appendix A.

### 4.9. Mass Spectrometry Data Analysis

The normalized phospho-proteomics intensities were log2-transformed. The proteins and corresponding genes showing significant differences among each pair of conditions were defined as those with at least one phosphorylation site showing a statistically significant difference (adjusted *p* < 0.05) and either an increase or a decrease of 20% in the phosphorylation level. The principal component analysis was performed on the entire set of phospho-sites. The analysis revealed a clear separation of the MR and MR-Axl cells. To enhance precision, the analysis was refined by focusing on the top 3000 most variable sites, taking parameters such as the Fano factor and the establishment of a minimum average intensity threshold into account. The clustering of the significantly changed phospho-sites was performed using k-means clustering from the ComplexHeatmap package (v2.6.2; [59]) with *k* = 4. This number of *k* was empirically optimized by inspecting the dendrogram obtained by hierarchical clustering, using the same parameters (Euclidean distance and complete linkage). The enrichr function from the clusterProfiler package (v 3.18.1; [60]) was used to perform the gene set enrichment analyses. Only those gene sets with at least two genes corresponding to the considered input set of differentially phosphorylated proteins were retained. Only the gene sets with an adjusted *p*-value < 0.05 in at least one comparison were retained for further consideration and analyses. The MSigDB Hallmark Gene Sets and KEGG Pathways were retrieved via the msigdbr package (v7.2.1). The bubble-plots were generated using ggplot2 (v 3.4.1). The PhosphoSite.mouse dataset from the PhosR package (v1.0.0; [25]) was then used as the reference annotation. A direction analysis was performed using the directPA package (v 1.5; [26]) separately for each direction. Direction n = 1 highlighted those kinases, or targets thereof, that are preferentially activated in Gas6-stimulated MR-Axl cells compared to Gas6-stimulated MR cells. Direction n = −1 highlighted instead those preferentially activated in Gas6-stimulated MR cells. The results were plotted using the perturbPlot2d function. These steps were all performed using the R statistical environment v4.0.4.

### 4.10. Western Blotting

Western blotting was performed as described previously [61]. The primary anti-Axl (1:2500; R&D, Boston, MA, USA; #AF154), anti-pAkt Ser 473 (1:500; Cell Signaling, Danvers, MA, USA; #4060), anti-Akt (1:500; Cell Signaling #4685) and anti-actin (1:2500; Sigma, Singapore; #A2066) antibodies were diluted in 5% BSA TBS-T (0.1% Tween) and incubated at 4 °C overnight with gentle agitation. The intensities of the individual proteins were assessed by densitometry for further normalization.

### 4.11. Cytokine Array

For the detection of cytokines in Axl^−/−^ and Axl^+/+^ tumors, the mouse inflammation antibody array (Abcam; #ab133999) was used according to the manufacturer’s protocol. In brief, the bulk tumor tissue (*n* = 4 tumors per group) was lysed using the buffer provided by the manufacturer. The antibody-spotted membranes were blocked for 1 h at room temperature. The cell lysates containing 250 µg protein were subjected to each membrane and incubated overnight at 4 °C with slight agitation. After incubation, the membranes were washed and incubated with biotinylated antibodies and HRP-labeled conjugates. After washing, the membranes were overlaid with detection buffer and exposed to X-ray films for different timepoints. The spots were analyzed using ImageJ version 1.38 (ImageJ, Bethesda, MD, USA) and the spot intensities were normalized to control spots on the respective membrane.

### 4.12. Animal Experiments

The animal husbandry and animal procedures were approved according to the Austrian guidelines for animal care and protection (BMBWF-66.009/0233-V/3b/2019). All the experiments were conducted in agreement with these protocols. For the endogenous mouse model of liver tumorigenesis, the Albumin-Cre expressing mice [62] were bred with mice having floxed Axl alleles (Axl^fl/fl^) to delete Axl specifically in hepatocytes (thereof Axl^Δhep^ mice, [41]). To study the systemic effects of Axl, we used Axl^−/−^ and Axl wild-type mice (Axl^+/+^) [63]. To induce endogenous liver tumor formation, diethylnitrosamine (DEN) (25 mg/kg) or the vehicle (0.9% NaCl) were injected intraperitoneally (i.p.) into 14-day-old Axl^−/−^ and Axl^+/+^ mice or Axl^Δhep^ or Axl^fl/fl^. At five weeks of age, the mice were injected weekly with CCl_4_ (0.5 µL/g) or the vehicle (corn oil) for 20 weeks to induce liver fibrosis and tumor formation in a fibrotic background. After the last CCl_4_ injection (22 weeks after DEN injection), the mice were sacrificed and the livers harvested for the subsequent comparative analysis. All the mice were maintained on a C57BL/6 background. To investigate the primary tumor burden and metastasis, we subcutaneously engrafted 5 × 10^6^ MR, MR-Axl, Hep3B-GFP or Hep3B-Axl cells into immunocompromised SCID mice. The tumors were measured with a caliper and the tumor volume was calculated using the formula [(length × width^2^)/2]. The MR and MR-Axl tumors were allowed to establish until the mean tumor volume of 50 mm^3^ was reached and then treated i.p. with either an Axl decoy receptor (20 mg/kg) or vehicle (PBS) [64]. The tumors were resected, treatment was discontinued and survival analysis based on tumor recurrence was evaluated. To study the blood vessel exit and metastatic colonization of the cells, 1 × 10^4^ MR or MR-Axl was injected into the lateral tail vein of the SCID mice. The mice were sacrificed 3 weeks after injection.

### 4.13. Immunohistochemistry

The tissues were fixed with 4% phosphate-buffered formaldehyde (PFA) for 24 h and embedded in paraffin. The 5 µm thick sections of liver and lung were stained with hematoxylin and eosin (H&E). The tumor burden was determined by counting the tumor nodules per liver or lung lobe and by calculating the ratio of tumor to liver or lung area. To quantify the collagen deposition and fibrosis in the liver, sections were stained with Sirius Red (0.1% Sirius Red in picric acid). The Sirius Red^+^ areas were quantified and normalized to the whole tissue area using Definiens Tissue Studio^®^ XD 64 software version 2.7 (Definiens Inc., Carlsbad, CA, USA). For the immunohistochemistry (IHC), 5 µm thick sections were stained using standard IHC protocols with antibodies against the following: Axl (1:100; R&D; #AF154), mouse Axl (1:100; R&D; #AF854), BrdU (1:1000; Sigma; #B2531), CD31 (1:200; Abcam; ab28364), CD8 (1:200; Cell Signaling; #98941), CD3 (1:200; Epredia^TM^; # 12644527), Granzyme B (1:100; Abcam; #ab4059), PD-L1 (1:200; Cell Signaling, #64988), PCNA (1:500; Dako; #M0879), E-cadherin (1:200; BD Biosciences; #610182) and vimentin (1:5000; Sigma; #V5255). 3,3′-Diaminobenzidine (DAB) was used as a chromogen. The DAB-positive cells were counted and normalized to DAB-positive and -negative cells to receive the relative positive cell count as indicated in the figure axis labels. The stained sections were quantified with Definiens Tissue Studio^®^ XD 64 software version 2.7 (Definiens Inc., Carlsbad, CA, USA).

### 4.14. Multiplex IHC

Tissue sections of the HCC patients were stained against Axl (1:50; R&D; #AF154), CD8 (1:200; Abcam; #ab4055), Granzyme B (1:100; Abcam; #ab4059), CD31 (1:100; Abcam; #ab28364), α-smooth muscle actin (1:100; Dako; #M0851), CD45 (1:100; BioLegend, San Diego, CA, USA; #B368502) and DAPI using the Opal Polaris 7 Color IHC Detection Kit (Akoya Biosciences; Marlborough, MA, USA). The stained sections were scanned using the Vectra Polaris^TM^ automated imaging system (Perkin Elmer; Hopkinton, MA; USA) and analyzed using HALO AI software version 3.6 (Indica labs; Albuquerque, NM; USA). The cells were classified according to their expression of DAPI and the respective marker. The relative cell count was obtained by normalizing the number of cells positive for both DAPI and the marker to the total number of cells positive for DAPI.

### 4.15. Flow Cytometry

A transcardial perfusion was performed with 20 mL PBS and the livers and spleens were harvested to generate single cell suspensions. The livers were minced to approximately 1 mm^3^ pieces and incubated in RPMI-1640 supplemented with collagenase type IV (2 mg/mL) and DNAse I (50 U/mL) for 1 h at 37 °C. The suspensions were filtered through 70 µm cell strainers and centrifuged at 100× *g* at 4 °C. The supernatants, including immune cells, were then filtered through 40 µm cell strainers and erythrocyte lysis was performed using the ACK lysis buffer (Life Technologies; Carsbad, USA). The cells were stained with anti-CD45 (1:100; APC/Fire^TM^ 750; BioLegend; #103153), anti-CD3 (1:100; redFluor™ 710; Cytek Biosciences, Fremont, CA, USA; #80-0032), anti-CD4 (1:100; FITC; Cytek Biosciences; #35-0042), anti-CD8 (1:200; PE; Cytek Biosciences; #50-0081), anti-PD-1 (1:50; Brilliant Violet 421; BioLegend; #135221), anti-CD19 (1:50; Brilliant Violet 421; #115537), anti-NK1.1 (1:100; PE/Cyanine7; BioLegend; #108713), anti-FoxP3 (1:50; APC; BioLegend; #15330750), anti-Ly6G (1:400; FITC; Cytek Biosciences; #35-1276), anti-Ly6C (1:400; PE/Cyanine7; BioLegend; #128017), anti-CD103 (1:50; APC; BioLegend; #121413), anti-MHCII (1:100; Brilliant Violet 785; BioLegend; #107645), anti-F4/80 (1:100; PE; Cytek Biosciences; # 50-4801) and anti-CD206 (1:50; Alexa Fluor ^®^ 700; BioLegend; #141733) antibodies. The live/dead cell staining was performed using 7-AAD or AquaZombie fixable dye^TM^ (BioLegend, San Diego, CA, USA) to exclude dead cells for the analysis. Flow cytometry was conducted using the LSR Fortessa^TM^ X20 cell analyzer (BD Biosciences, Franklin Lakes, NJ, USA). The data were analyzed using FlowJo (BD Biosciences, Franklin Lakes, NJ, USA).

### 4.16. Statistical Analysis

For the statistical analysis, we used GraphPad Prism 5 (GraphPad Software Inc, San Diego, CA, USA). The data are expressed as the mean ± standard deviation (SD). For the comparison of two groups, the Student’s *t*-test was used, except for the Kaplan–Meier analysis, where a log-rank test was applied. For the correlation analysis of two groups, the Pearson correlation co-efficient was determined. For the comparison of more than two groups, the Oneway ANOVA followed by Tukey’s multiple comparison test was employed. Grubbs’s test was used for the detection of outliers. Statistically significant *p*-values were considered and indicated in graphs as * *p* < 0.05; ** *p* < 0.01; and *** *p* < 0.001.

## 5. Conclusions

Together, these data show that tumor-intrinsic Axl expression fosters progression via a partial EMT, while tumor-extrinsic Axl expression shapes an inflammatory microenvironment that curbs tumor development.

## Figures and Tables

**Figure 1 ijms-25-04202-f001:**
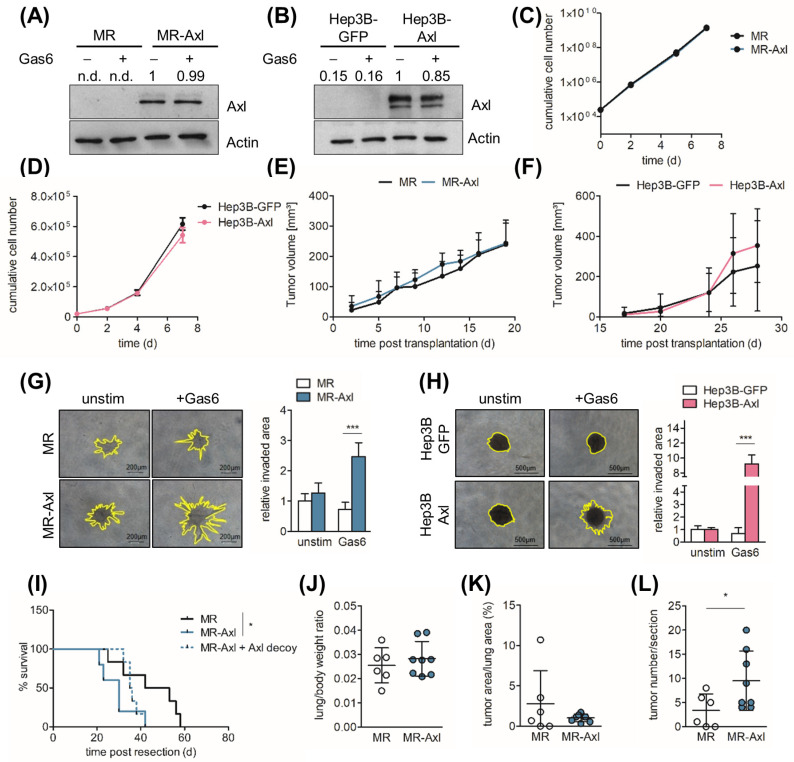
Axl promotes invasion of malignant hepatocytes. (**A**,**B**) Axl expression of MR, MR-Axl, Hep3B-GFP and Hep3B-Axl cells as determined by Western blotting. Actin expression is shown as loading control. Signal intensities were assessed by densitometry. The expression of Axl was normalized to actin levels. Normalized Axl expression in unstimulated MR-Axl and Hep3B-Axl cells was set to the value of 1. (**C**,**D**) Cumulative numbers of MR, MR-Axl, Hep3B-GFP and Hep3B-Axl cells. (**E**) Tumor volume of MR and MR-Axl tumors after subcutaneous injection into SCID mice (*n =* 6 per group). (**F**) Tumor volume of subcutaneous Hep3B-GFP (*n* = 3) and Hep3B-Axl-derived tumors (*n* = 6) in SCID mice. (**G**,**H**) Invasion of tumor spheroids generated from MR, MR-Axl, Hep3B-GFP and Hep3B-Axl cells. Spheroids were treated with Gas6 (500 ng/mL) or vehicle for 24 h. Quantifications show invaded areas relative to unstimulated MR and Hep3B-GFP cells. The invasive area of MR and Hep3B tumor spheroids was set to a value of 1. (**I**) Kaplan–Meier plot of SCID mice after resection of subcutaneous MR (*n* = 6), MR-Axl tumors (*n* = 4) and MR-Axl tumors treated with Axl decoy receptor (*n* = 6). Tumors were resected 19 days after injection. (**J**) Lung/body weight ratio, (**K**) tumor area/lung area and (**L**) tumor number/section of mice intravenously injected with MR (*n* = 6) and MR-Axl (*n =* 8) cells. N.d., not detected. Data are expressed as mean ± SD. *. *p* < 0.05; ***. *p* < 0.001.

**Figure 2 ijms-25-04202-f002:**
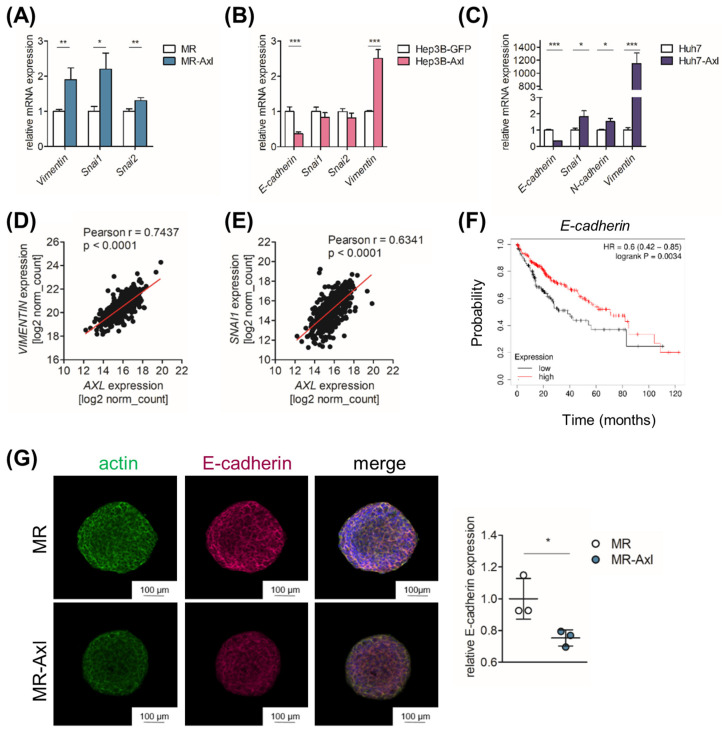
Axl is associated with EMT in liver cancer cells. (**A**–**C**) Relative mRNA expression of EMT-associated genes in (**A**) MR and MR-Axl cells, (**B**) Hep3B-GFP and Hep3B-Axl cells and (**C**) Huh7 and Huh7-Axl cells as determined by qPCR. Expression of MR, Hep3B-GFP or Huh7 cells was set to the value of 1. (**D**,**E**) Correlation of (**D**) *Vimentin* and *Axl* expression or (**E**) *Snai1* and *Axl* of HCC samples retrieved from TCGA LIHC data (*n =* 371). (**F**) Kaplan–Meier curve showing HCC patients with high (red) and low (black) *E-Cadherin* expression (*n =* 364). (**G**) Immunofluorescence analysis of MR and MR-Axl spheroids stimulated with Gas6 (500 ng/mL) for 24 h. Spheroids were stained for E-cadherin (red), actin (green) and DAPI (blue). Dot plots depict quantification of E-cadherin expression of Gas6-stimulated spheroids. E-Cadherin expression was normalized to actin expression and further normalized to MR spheroids, which was set to the value of 1. *. *p* < 0.05; **. *p* < 0.01; ***. *p* < 0.001.

**Figure 3 ijms-25-04202-f003:**
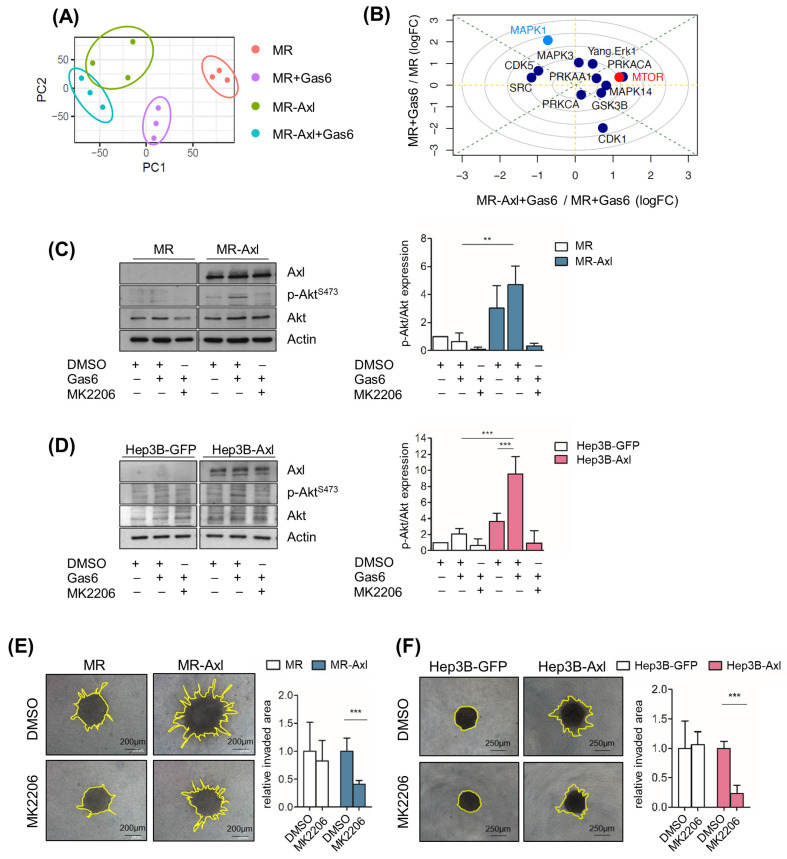
Invasion of liver cancer cells depends on Axl/Akt signaling. (**A**) Principal component (PC) analysis showing sample-dependent clustering of MR, MR-Axl, MR + Gas6 and MR-Axl + Gas6 cells. (**B**) Kinase perturbation analysis using PhosR. Scatterplot highlights the change in activity for the indicated kinases in MR + Gas6 (y-axis; using the corresponding untreated cells as baseline) vs. MR-Axl + Gas6 (x-axis). Those kinases, or targets thereof, that are preferentially activated in MR-Axl + Gas6 are highlighted in red. Those kinases, or targets thereof, preferentially activated in MR + Gas6 are highlighted in light blue. Dark blue circles indicate non-significantly regulated kinases. (**C**) Western blot analysis of MR and MR-Axl cells (*n =* 3). Cells were either untreated or treated with Gas6 (500 ng/mL) and MK2206 (1.25 µM) for 15 min. Actin expression is shown as loading control (left panel). Signal intensities were assessed by densitometry. p-Akt levels were normalized to actin and total Akt expression. p-Akt signal of unstimulated MR cells was set to the value of 1 (right panel). (**D**) Western blot analysis of Hep3B-GFP and Hep3B-Axl cells (*n =* 3). Cells were either untreated or treated with Gas6 (500 ng/mL) and MK2206 (1.25 µM) for 15 min (left panel). Normalization of p-Akt was performed as described in 3C (right panel). p-Akt signal of unstimulated Hep3B-GFP cells was set to the value of 1. (**E**,**F**) Invasion of MR-Axl (**E**) and Hep3B-Axl (**F**) spheroids after stimulation with Gas6 (500 ng/mL) and inhibition of Akt signaling with MK2206 (1.25 µM). Gas6 and MK2206 were added into collagen and media 24 h prior to imaging. Bar charts depict invaded area relative to DMSO control. Invasive area of DMSO control spheroids was set to a value of 1. Data are expressed as mean ± SD. **: *p* < 0.01; ***: *p* < 0.001.

**Figure 4 ijms-25-04202-f004:**
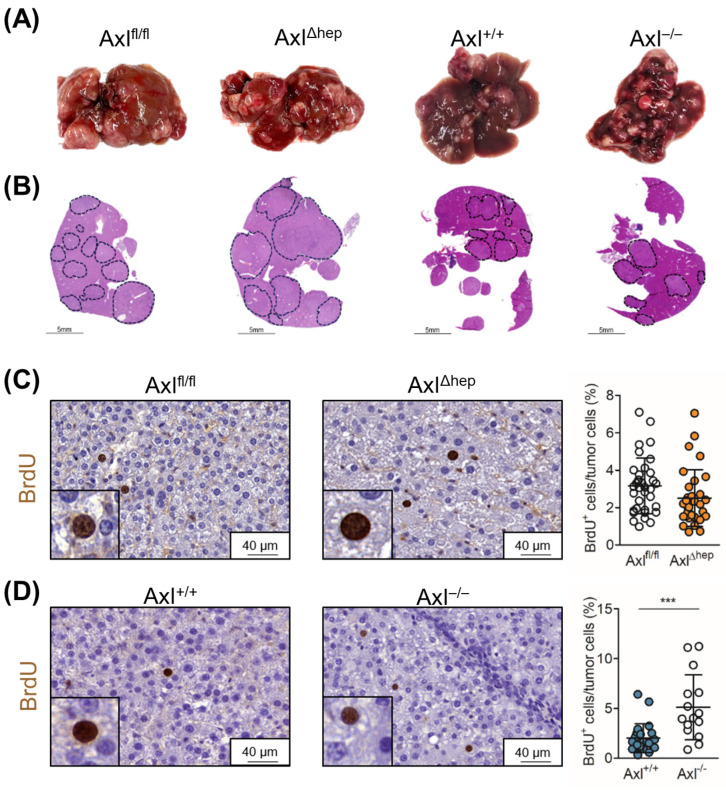
Enhanced proliferation of Axl^−/−^ liver tumors. (**A**) Representative images and (**B**) H&E staining of DEN+CCl_4_-induced liver tumors of Axl^fl/fl^, Axl^Δhep^, Axl^+/+^ and Axl^−/−^ mice. Dotted lines indicate tumor nodules. (**C**) Representative images of BrdU^+^ cells of Axl^fl/fl^ (white) and Axl^Δhep^ (orange) tumors as determined by immunohistochemistry. Quantification depicts BrdU^+^ cells relative to tumor cells. Three tumors per mouse were used for the analysis. (**D**) Analysis of BrdU^+^ cells in Axl^+/+^ (blue) and Axl^−/−^ (white) tumors as described in (**C**). Data are expressed as mean ± SD. ***: *p* < 0.001.

**Figure 5 ijms-25-04202-f005:**
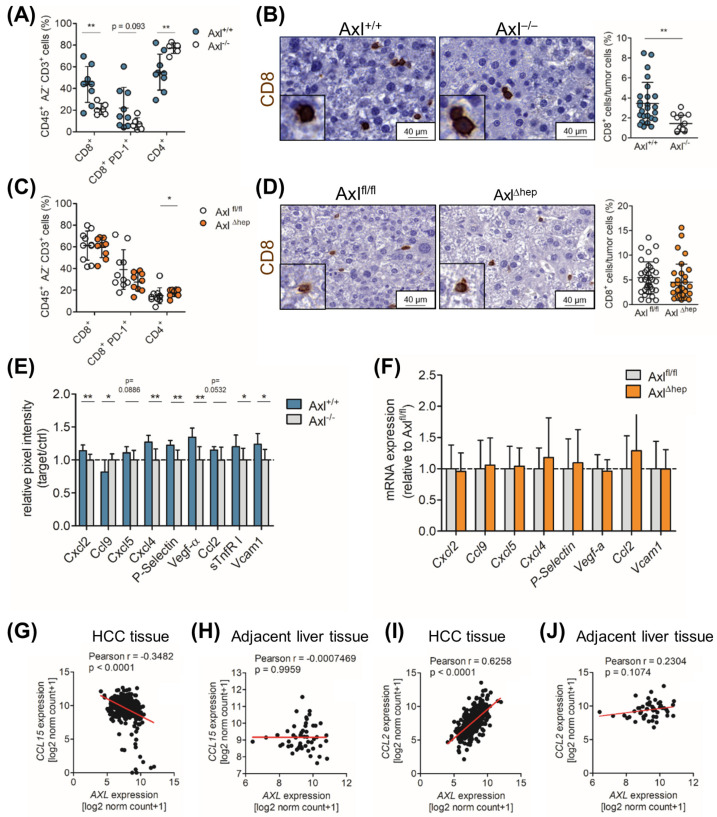
Axl expression stimulates infiltration of cytotoxic immune cells. (**A**) Flow cytometry analysis showing the relative abundance of CD8^+^, CD8^+^ PD-1^+^ and CD4^+^ T cells in liver tumors of Axl^+/+^ (*n =* 9) and Axl^−/−^ (*n =* 6) mice. (**B**) Representative images of CD8^+^ cells in Axl^+/+^ and Axl^−/−^ liver tumors as determined by immunohistochemistry. Quantification depicts CD8^+^ cells relative to tumor cells. Three tumors per mouse were analyzed. (**C**) Flow cytometry analysis of tumors from Axl^fl/fl^ (*n =* 10) and Axl^Δhep^ (*n =* 9) mice as described in (**A**). (**D**) Representative images of CD8+ cells of Axl^fl/fl^ and Axl^Δhep^ tumors as detected by immunohistochemistry. Quantification depicts CD8^+^ cells relative to all tumor cells. Three tumors per mouse were analyzed. (**E**) Quantification of cytokine arrays using Axl^+/+^ and Axl^−/−^ tumor tissue lysates (*n =* 4 mice per group). Spot intensities were normalized to control on each membrane and normalized to Axl^−/−^ samples. Normalized spot intensities of Axl^−/−^ tumor tissue were set to a value of 1. (**F**) Transcript levels of *Cxcl2, Ccl9, Cxcl5, Cxcl4, P-selectin, Vegf-a, Ccl2* and *Vcam1* of tumor tissue lysates of Axl^fl/fl^ and Axl^Δhep^ mice as determined by qPCR. The expression is shown relative to Axl^fl/fl^ tumors, which was set to the value of 1. (**G**) Correlation of *CCL15* and *Axl* expression of HCC samples retrieved from TCGA LIHC data (*n =* 371). (**H**) Correlation of *CCL15* and *Axl* expression of normal adjacent liver tissue samples retrieved from TCGA LIHC data (*n =* 50). (**I**) Correlation analysis of *CCL2* and *Axl* expression of HCC samples retrieved from TCGA LIHC data (*n =* 371). (**J**) Correlation analysis of *CCL2* and *Axl* expression of normal adjacent liver tissue samples retrieved from TCGA LIHC data (*n =* 50). Regression lines are shown in red (**G**–**J**). Data are expressed as mean ± SD. *. *p* < 0.05; **. *p* < 0.01.

**Figure 6 ijms-25-04202-f006:**
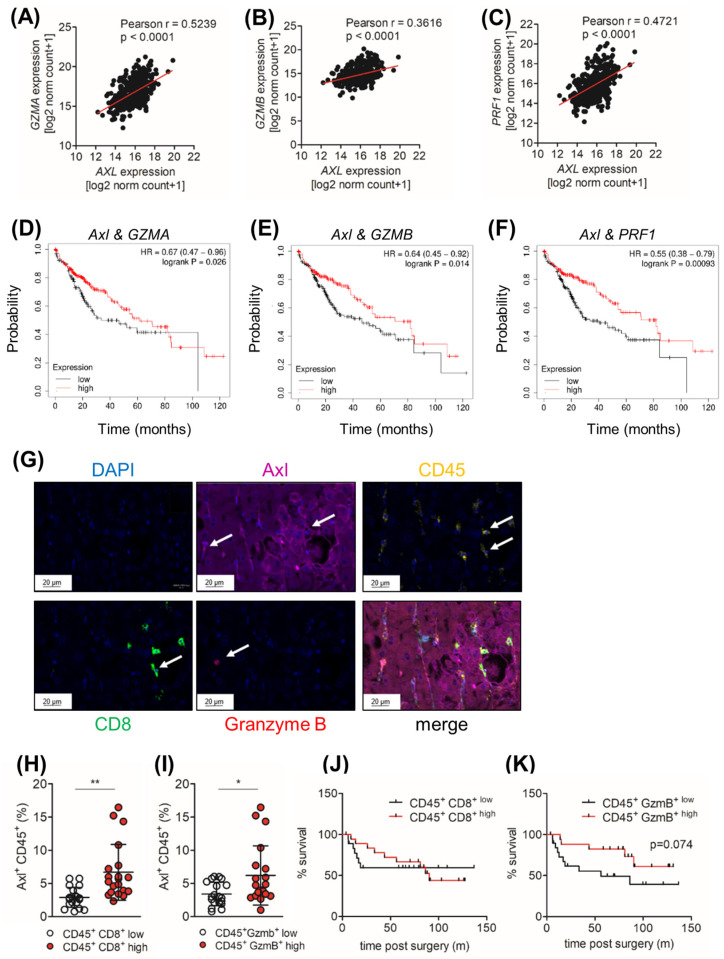
The infiltration of CD8+ and Granzyme B+ cells correlates with Axl expression on CD45+ cells in HCC patients. (**A**) A correlation analysis of *Granzyme A (GZMA),* (**B**) *Granzyme B (GZMB)* and (**C**) *Perforin-1(PRF1)* expression with *Axl* expression of HCC samples retrieved from the TCGA LIHC data (*n =* 371). Regression lines are shown in red. (**D**–**F**) A Kaplan–Meier curve showing the survival probability of HCC patients with high (red) and low (black) *Axl* and (**D**) *Granzyme A (GZMA)*, (**E**) *Granzyme B (GZMB)* and (**F**) *Perforin-1 (PRF1)* expression (*n =* 364). (**G**) Representative pictures of multiplexed immunohistochemical stainings of DAPI (blue), Axl (pink), CD45 (orange), CD8 (green) and Granzyme B (red). White arrows indicate cells positive for respective marker. (**H**) Quantification of CD45^+^ CD8^+ low^ (white) and CD45^+^ CD8^+ high^ (red) from Axl^+^ CD45^+^ cells of HCC patients (*n =* 40). (**I**) Quantification of CD45^+^ Granzyme B^+ low^ (white) and CD45^+^ Granzyme B^+ high^ (red) from Axl^+^ CD45^+^ cells of HCC patients (*n =* 40). (**J**) A Kaplan–Meier curve showing the survival probability of HCC patients with a high (red) and low (black) infiltration of CD45^+^ CD8^+^ T cells (*n =* 40). (**K**) A Kaplan–Meier curve showing the survival probability of HCC patients with a high (red) and low (black) infiltration of CD45^+^ Granzyme B^+^ cells (*n =* 40). The data are expressed as mean ± SD. *. *p* < 0.05; **. *p* < 0.01.

## Data Availability

The mass spectrometry proteomics data are openly available in the ProteomeXchange Consortium via the PRIDE partner repository with the dataset identifier PXD047577 [65].

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
