# Peer review of "Tumor-Extrinsic Axl Expression Shapes an Inflammatory Microenvironment Independent of Tumor Cell Promoting Axl Signaling in Hepatocellular Carcinoma"

_ijms, 2024, doi:10.3390/ijms25084202_

Round 1

Reviewer 1 Report

Comments and Suggestions for Authors

In this manuscript, Breitenecker observed that tumor intrinsic Axl and extrinsic Axl play a different role in liver cancer.  The author showed that tumor-intrinsic Axl expression helps progression while tumor-extrinsic Axl expression controls an inflammatory tumor microenvironment. The study presents intriguing data, but many key data are not included in the package. 

Concerns are listed below.

The supplementary file only contains “Main Figures” and does not include “Supplementary Figures”.

The Introduction section only mentioned AXL's negative role, such as inducing PDL1 expression, reducing pro-inflammatory cytokine, fibrosis, etc.  Including references that highlight the positive roles of AXL can help bolster the rationale by presenting a more comprehensive view of its functions.

It appears that some of the MR-AXL mice die after tumor resection. Therefore, it may be beneficial to extend the tumor burden experiment depicted in Figure 1E and include a survival curve analysis.  FigS1G should be incorporated in Fig1I. Additionally, it is advisable to include the log-rank test in Figure 1I for a more comprehensive statistical analysis.

To study the systemic effect of Axl, the author used Axl-/- mice.  However, it is not clear why Axl (-/-) mice still express Axl in stromal. FigS4K is not provided.  

Line 236-The author indicates that tumor-cell extrinsic Axl expression increases the proliferation of hepatocellular tumor cells. However, Line 246-The author introduced the idea that extrinsic Axl expression reduces tumor cell proliferation. This is confusing. 

The consensus is that high expression of the Axl receptor is linked to a poorer prognosis in liver cancer. However, the data presented in this study seems to suggest that Axl expression may benefit immunotherapy. While this is indeed intriguing, the data provided is not entirely clear, and more detailed information is needed to better understand this phenomenon.

Reviewer 2 Report

Comments and Suggestions for Authors

Breitenecker et al. present an investigative study into the roles of Axl receptor tyrosine kinase in the context of hepatocellular carcinoma (HCC), particularly differentiating between its tumor-intrinsic and tumor-extrinsic effects. Utilizing both murine models and analyses of patient data, the authors elucidate the non-cell-autonomous functions of Axl within the tumor microenvironment (TME). This research contributes to the understanding of complex TME dynamics and the multifaceted role of Axl in HCC progression.

2. General Evaluation:

The study is well-conceived and addresses a significant gap in the understanding of Axl's role in HCC. The dual approach combining animal models with patient sample analyses strengthens the study's validity. However, to enhance the manuscript's scientific merit and clarity, several areas require further attention and refinement.

3. Specific Comments:

Introduction:

  • The introduction provides an adequate background on HCC and Axl's involvement. It could be enhanced by a more detailed review of Axl's known biological functions and its implications in other cancers, setting a comprehensive stage for the current study.

Materials and Methods:

  • The description of methodologies is generally clear but lacks certain specifics, particularly in patient sample selection and the bioinformatics pipeline. Detailed inclusion criteria and analysis parameters should be provided for reproducibility.

Results:

  • Results are presented in a structured manner, contributing valuable insights into Axl's differential roles. A more detailed exploration of the underlying mechanisms through which Axl modulates the TME would strengthen this section.
  • Supplementary data or additional experimental validations would bolster the findings related to tumor-extrinsic Axl expressions.

Discussion:

  • This section adeptly contextualizes the findings within existing literature. However, it could be broadened to explore the broader implications of the differential roles of Axl in therapeutic strategies against HCC.
  • Discussion of the study's limitations, including model-specific biases or limitations of the dataset, would provide a more comprehensive view.

Figures and Tables:

  • Figures provide relevant information but require improvements in resolution and labeling for better clarity and interpretability.
  • Consistency in data presentation across all visual elements should be ensured, including uniformity in legends and figure captions.

Technical and Language Aspects:

  • The manuscript would benefit from thorough proofreading to address typographical and grammatical errors, enhancing overall readability and professionalism.
  • Ensure that the formatting, especially in the references section and figure captions, adheres to the journal's guidelines.

Reviewer 3 Report

Comments and Suggestions for Authors

This study aims to elucidate the tumor-cell intrinsic and extrinsic roles of Axl in hepatocellular carcinoma (HCC) and mouse models. The authors discovered that tumor-cell intrinsic expression of Axl promotes cell invasiveness and induces partial epithelial-mesenchymal transition (EMT), while tumor-cell extrinsic expression of Axl increases the infiltration of immune cells. These findings suggest that targeting Axl could be effective in treating advanced HCC.

The background of the study is well described, providing readers with a comprehensive understanding of the significance of HCC and Axl. Specifically, the scientific basis for Axl's involvement in EMT and the regulation of the immune microenvironment is clearly presented. The results are articulated clearly with appropriate experimental data, offering a logical interpretation and an easy-to-follow conclusion path. The discussion extensively covers the scientific significance of the results, offering a detailed analysis of how the study's conclusions relate to the current literature. The impact of the study is significant, as it reveals potential benefits of targeting Axl in the treatment of HCC, making it of great interest to clinicians and researchers seeking treatment strategies for HCC, as well as immunologists interested in the role of Axl in the regulation of the tumor microenvironment. The relationship between Axl and immune cell infiltration could provide new directions in cancer immunotherapy. I recommend this paper for acceptance, noting that with the following modifications, it can appeal to a broader audience and further enhance its impact.

Recommendations for Improvement

  1. To target a wider reader base, it would be beneficial to add a brief introduction to Axl, EMT, and HCC at the beginning of the introduction section. This would not only cater to specialists but also make the paper accessible to a broader scientific community, including those not immediately familiar with the field.

In conclusion, this paper demonstrates the significant role of Axl in the progression and treatment of HCC, contributing valuable insights into the field. With the suggested modifications, the paper will likely achieve greater reach and impact.

Round 2

Reviewer 1 Report

Comments and Suggestions for Authors

In this resubmission, the author has provided additional evidence to justify the dichotomous role of Axl in liver cancer cells. The authors demonstrated that intrinsic-Axl promotes tumor cell invasion and metastasis to the lung, while extrinsic-Axl modulates immune effector cells in the tumor microenvironment (TME). The inclusion of supplementary data further supports the extrinsic role of Axl in liver cancer.  The utilization of Axlfl/fl, AxlΔhep, Axl+/+, and Axl-/- mouse models is well-designed, and the outcomes now provide a clear intrinsic/extrinsic role of Axl. Additionally, the study reveals interesting results indicating that CD8+ and Granzyme B+ cells are highly infiltrated in Axl+/+ tumors compared to Axl-/- tumors.

Suggesting the straightforward in vitro co-culture experiment with human-PBMCs or mouse-splenocytes and specific subtypes of Axl+ and Axl-cell lines could provide further insights into Axl's role in reprogramming immune cells. The author acknowledges this experiment's potential and mentions it as a limitation of the study. It would be beneficial if the author decided to include this experiment in the manuscript. Nevertheless, the current information provided is sufficient to convince the intrinsic/extrinsic role of Axl in liver cancer.